# Biochemical patterns of antibody polyreactivity revealed through a bioinformatics-based analysis of CDR loops

Christopher T Boughter[1], Marta T Borowska[2], Jenna J Guthmiller[3], Albert Bendelac[4,5], Patrick C Wilson[3,4], Benoit Roux[2], Erin J Adams[2,4]*

[1]Graduate Program in Biophysical Sciences, University of Chicago, Chicago, United States; [2]Department of Biochemistry and Molecular Biology, University of Chicago, Chicago, United States; [3]Department of Medicine, Section of Rheumatology, University of Chicago, Chicago, United States; [4]Committee on Immunology, University of Chicago, Chicago, United States; [5]Department of Pathology, University of Chicago, Chicago, United States

**Abstract** Antibodies are critical components of adaptive immunity, binding with high affinity to pathogenic epitopes. Antibodies undergo rigorous selection to achieve this high affinity, yet some maintain an additional basal level of low affinity, broad reactivity to diverse epitopes, a phenomenon termed 'polyreactivity'. While polyreactivity has been observed in antibodies isolated from various immunological niches, the biophysical properties that allow for promiscuity in a protein selected for high-affinity binding to a single target remain unclear. Using a database of over 1000 polyreactive and non-polyreactive antibody sequences, we created a bioinformatic pipeline to isolate key determinants of polyreactivity. These determinants, which include an increase in inter-loop crosstalk and a propensity for a neutral binding surface, are sufficient to generate a classifier able to identify polyreactive antibodies with over 75% accuracy. The framework from which this classifier was built is generalizable, and represents a powerful, automated pipeline for future immune repertoire analysis.

*For correspondence:
ejadams@uchicago.edu

Competing interests: The authors declare that no competing interests exist.

## Introduction

Antibodies are immunogenic proteins expressed by B cells that play a major role in the adaptive immune response against non-self. Upon recognition of target epitopes, these antibodies undergo multiple rounds of somatic hypermutation and affinity maturation inside a germinal center, whereby the amino acid sequence of the epitope-binding surface is selected for optimal binding to the target (*Victora and Nussenzweig, 2012*; *Eisen and Siskind, 1964*; *McKean et al., 1984*). The longer this affinity maturation process extends, the higher the affinity and specificity of the antibodies toward their target antigen, primarily through mutagenesis of the six complementarity determining region (CDR) loops of the antibody (*Victora and Nussenzweig, 2012*). Using a combination of affinity matured CDR loops, these antibodies bind strongly to the target and aid in invader neutralization. While the process of affinity maturation and somatic hypermutation of antibodies results in high-affinity and incredibly specific binders to a particular epitope, some antibodies have been shown to display signs of reactivity toward diverse off-target epitopes. This broad but low-affinity binding has been termed 'polyreactivity'.

Antibody polyreactivity has been hypothesized to be beneficial in the early stages of antibody maturation, acting as a pool of diverse binders ready to recognize novel antigens and initiate the

**eLife digest** To defend itself against bacteria and viruses, the body depends on a group of proteins known as antibodies. Each subset of antibodies undergoes a rigorous training regimen to ensure it recognizes a single epitope well – that is, one specific region on the surface of foreign, harmful organisms.

Most antibodies stick extremely tightly to their one unique epitope, but some can also weakly bind to molecules that are vastly different from their main trained targets. This feature – known as polyreactivity – can in some cases help the immune system fight against multiple strains of viruses. On the other hand, when antibodies are designed in the laboratory to treat diseases, this characteristic can sometimes lead to the failure of pre-clinical trials. Yet it is currently unclear why some antibodies are polyreactive when others are not.

To investigate this question, Boughter et al. compared over 1,000 polyreactive and non-polyreactive antibody sequences from a large database, revealing differences in the physical properties of the region of the antibodies that attaches to epitopes. Using these defining features, Boughter et al. went on to design a new piece of freely available, automated software that could predict which antibodies would be polyreactive more than 75% of the time.

Such software could ultimately help to guide the design of antibody-based treatments, while bypassing the need for costly laboratory tests.

more stringent selection process (*Dimitrov et al., 2013*). To this end, a majority of B cell receptors and antibodies which have not undergone somatic hypermutation, including those on immature B cells and early 'natural' antibodies, have been found to be polyreactive to some extent and are suggested to have an innate-like response to pathogens (*Ochsenbein et al., 1999*; *Wardemann et al., 2003*). While these mostly unmutated polyreactive antibodies remain at low frequency in antigen-experienced individuals, a distinct population of polyreactive antibodies that have undergone selection are still expressed by mature B cells that circulate in blood (*Tiller et al., 2007*). In fact, some studies have found that the polyreactivity status of an antibody is mostly independent of the number of somatic hypermutations in the antibody sequence (*Mouquet et al., 2010*; *Prigent et al., 2016*). In line with this finding, only 5–10% of the repertoire of naive B cells circulating in the periphery are polyreactive, but this increases to 20–30% in the memory B cell compartment, showing a distinct capability of polyreactivity to survive selection (*Tiller et al., 2007*; *Koelsch et al., 2007*). These results suggest that polyreactivity can persist, or perhaps even be selected for during the selection process within the germinal center.

In a few notable cases, polyreactivity may in fact augment the efficacy of a given immune response. Polyreactive IgA antibodies have been shown to have an inherent reactivity to microbiota in the mouse gut, with a predicted role in host homeostasis (*Bunker et al., 2017*). These previously identified antibodies so far have no known primary ligands, yet play a key role in facilitating the gut immune response to the plethora of exogenous antigens encountered in the dynamic dietary and microbial environment of the gut. This implies the existence of antibodies whose primary function is to act as polyreactive sentries in the gut, yet the downstream effects of polyreactive antibodies coating commensal bacteria is so far unclear. Similar polyreactive IgA and IgG mucosal antibodies were found in the gut of human immunodeficiency virus (HIV)-infected patients, but these antibodies either had low affinity to the virus or lacked neutralization capabilities (*Planchais et al., 2019*). The benefit of singular antibody sequences with the ability to sample large portions of the commensal population may represent an improvement in efficiency of the homeostatic machinery of the gut.

While the precise role of these primarily polyreactive gut antibodies is still a topic of debate, polyreactivity has been suggested to augment the immune response in other immunological niches. Broadly neutralizing antibodies (bnAbs), which bind robustly to conserved epitopes on the surface glycoproteins of influenza viruses or HIV are more likely to be polyreactive (*Haynes et al., 2005*; *Mouquet et al., 2011*; *Andrews et al., 2015*). In one study of HIV-binding antibodies, over half of all tested bnAbs were found to be polyreactive (*Prigent et al., 2018*). These bnAbs have been the subject of intense study for their potential as the central components of an HIV treatment or as the byproduct of an immune response to a universal Influenza vaccine (*Andrews et al., 2015*;

*Haynes et al., 2019*; *Crowell et al., 2019*; *Li et al., 2012*). One hypothesized mechanism for the capability of polyreactive antibodies to confer this broad neutralization in the face of a changing viral epitope is heteroligation, the ability of a single antibody to bind the primary target with one binding domain and use the other binding domain to bind in a polyreactive manner (*Mouquet et al., 2010*). This heteroligation allows the antibody to take advantage of the significant avidity increase afforded by bivalent binding, despite the low envelope protein density of HIV or a geometry which does not readily lend itself to bivalent binding on the surface of influenza viruses (*Klein and Bjorkman, 2010*).

Although polyreactivity may play a positive role in natural immune responses, oftentimes this same property is considered undesirable from the point of view of generating therapeutic antibodies with high specificity. Antibody-based treatments, which generally take the form of an intravenous transfusion, are sensitive to the accelerated systemic clearance of polyreactive antibodies (*Hötzel et al., 2012*; *Kelly et al., 2015*; *Kelly et al., 2018*; *Datta-Mannan et al., 2015*). In general, much work has focused on attempting to answer the question of optimizing 'developability' of a given antibody. These efforts have been dedicated to determining the most critical components of developability through a large array of experimental assays, in silico structural prediction-based methods, sequence-based analysis and their correlations with clearance, sequence-based SASA predictions, and sequence-based aggregation propensity predictors (*Jain et al., 2017b*; *Raybould et al., 2019*; *Sharma et al., 2014*; *Jain et al., 2017a*; *Obrezanova et al., 2015*). In many of these studies, polyreactivity or non-specificity in general was seen to be a negative indicator of the developability of a drug, suggesting that therapeutic antibodies should strive toward a drug-like specificity (*Starr and Tessier, 2019*).

In line with this goal of understanding the predominant factors involved in the specificity of therapeutic antibodies, many researchers have worked to identify the biophysical underpinnings of polyreactivity in natural immune responses. The most popular hypotheses for the primary biophysical predictors of polyreactivity have included CDR3 length (*Prigent et al., 2016*), CDR3 flexibility (*Prigent et al., 2018*), net hydrophobicity (*Lecerf et al., 2019*), and net charge (*Rabia et al., 2018*). More observational studies have found an increased prevalence of arginine and tyrosine in polyreactive antibodies (*Kelly et al., 2018*; *Birtalan et al., 2008*). While these previous studies represent substantial advances in the study of polyreactivity, they have often been limited in scope, focusing on a singular antibody source and primarily focused on CDR3H. Comparing across these individual antibody sources highlights discrepancies between the proposed predictors of polyreactivity. The aforementioned properties determined to be key to polyreactivity in previous studies were found to be statistically insignificant in studies of HIV-binding and mouse gut polyreactive antibodies (*Bunker et al., 2017*; *Mouquet et al., 2010*).

Clearly, a computational framework that would enable us to predict the polyreactivity of a given antibody a priori, whether evaluating the efficacy of a natural immune response or the potential fate of a therapeutic antibody, would be tremendously useful. Such a framework, for example, could be used to assist in the isolation of broadly neutralizing antiviral antibodies, or speed up the process of therapeutic antibody screening. To achieve this goal, a thorough understanding of the molecular features behind polyreactive binding interactions is critical. Experimental approaches utilizing next-generation sequencing and ELISA allow for the identification of hundreds of polyreactive antibody sequences. However, the systematic characterization of these antibodies is difficult. Issues immediately arise when defining the conditions by which we determine an antibody to be polyreactive. While polyreactivity may exist on some continuous spectrum, we are inclined to frame the problem as binary. This binary discretization is useful for the identification of meaningful differences, yet must be recognized as an imperfect assumption. In addition to this more philosophical challenge, experimental efforts must also overcome significant hurdles. Detailed biochemical studies of polyreactive antibodies via protein crystallography, quantitative binding experiments, and mutagenesis provide exceptional insight but are inherently low throughput. Structural modeling of these polyreactive antibodies represent a high-throughput approach, but models of flexible loops are relatively unreliable, and are unlikely to capture nuances in side-chain placement (*Karami et al., 2019*). A bioinformatics-based approach, centered around high-throughput analysis that minimizes structural assumptions while maintaining positional context of amino acid sequences would provide a thorough, unbiased analysis of existing data and create a powerful pipeline for future studies.

In this study, we show that using just the amino acid sequences of antibodies from a database of over 1000 sequences tested for polyreactivity, unifying biophysical properties that distinguish

polyreactive antibodies from non-polyreactive antibodies can be identified. We find that, while charge and hydrophobicity are in fact important determinants of polyreactivity, the characteristic feature of polyreactive antibodies appears to be a shift toward neutrality of the binding interface. In addition, loop crosstalk is more prevalent in the heavy chain of polyreactive antibodies than non-polyreactive antibodies. From these properties, a machine learning-based classification software was developed with the capability to determine the polyreactivity status of a given sequence. This software is generalizable and can be retrained on any binary classification problem and identify the key differences between two distinct populations of antibodies, T cell receptors, or MHC-like molecules at the amino acid level.

## Results

### Database

Our aggregate database of over 1000 antibody sequences is compiled from our own previously published and new data, in addition to data from published studies by the Mouquet and Nussenzweig labs (*Table 1*; *Mouquet et al., 2010*; *Bunker et al., 2017*; *Planchais et al., 2019*; *Mouquet et al., 2011*; *Prigent et al., 2018*). Using an ELISA-based assay, the reactivity of each antibody is tested against a panel of 4–7 biochemically diverse target antigens: DNA, insulin, lipopolysaccharide (LPS), flagellin, albumin, cardiolipin, and keyhole limpet hemocyanin (KLH). This panel has become increasingly prevalent in the literature for experimental measures of polyreactivity in antibodies (*Mouquet et al., 2010*; *Prigent et al., 2016*; *Bunker et al., 2017*; *Planchais et al., 2019*; *Mouquet et al., 2011*; *Andrews et al., 2015*; *Prigent et al., 2018*; *Jain et al., 2017b*; *Neu et al., 2019*; *Wrammert et al., 2011*). The ligands represent a diverse sampling of biophysical and biochemical properties: for example, enrichment in negative charge (DNA, insulin, LPS, albumin), amphipathic in nature (LPS, cardiolipin), exceptionally polar (KLH), or large in size (KLH, flagellin). From this panel, a general rating of 'polyreactive' or 'non-polyreactive' is given to 529 and 524 antibodies, respectively. For the purposes of this study, antibodies are determined to be polyreactive if the authors of the original studies determined a particular clone binds to two or more ligands in the panel. Those that bind to one or none of the ligands in the panel are deemed non-polyreactive.

A limitation of this full polyreactivity dataset is that there exists an intermediate between the two classes. As discussed in the introduction, it is not immediately obvious where the line for polyreactivity should be drawn. An antibody that binds to 2–3 ligands may not necessarily achieve broad reactivity through the same mechanism as an antibody that binds four or more ligands from a panel of 6 or 7. To remove these ambiguities, a so-called 'parsed' dataset is generated whereby antibodies that bind 4–7 ligands are labeled polyreactive, antibodies that bind 0 panel ligands are labeled non-polyreactive, and those that bind 1–3 are removed from the analysis. The results presented below utilize the full dataset of 1053 antibody sequences, unless otherwise noted. Analysis of the parsed dataset can be found in the supplementary figures.

### A surface-level analysis of polyreactive antibody sequences

As a first pass at the given dataset, we focus on the most simplistic of the possible explanations for differences between polyreactive and non-polyreactive antibodies, specifically the J- and V-gene usage of each group. *Figure 1A and B*, rendered with code adapted from the Dash et al. derived program TCRdist (*Dash et al., 2017*), represents each antibody V-gene as a line connecting a single heavy and light chain gene for the full human-derived antibody dataset (685 sequences). We repeat this analysis in *Figure 1—figure supplement 1* using the parsed human-derived antibody dataset

**Table 1.** A quantification of the antibodies used in this study.

| Dataset | # Polyreactive | # Non-Polyreactive | Total |
|---|---|---|---|
| Mouse IgA | 205 | 240 | 445 |
| HIV reactive | 172 | 124 | 296 |
| Influenza reactive | 152 | 160 | 312 |
| Complete dataset | 529 | 524 | 1053 |

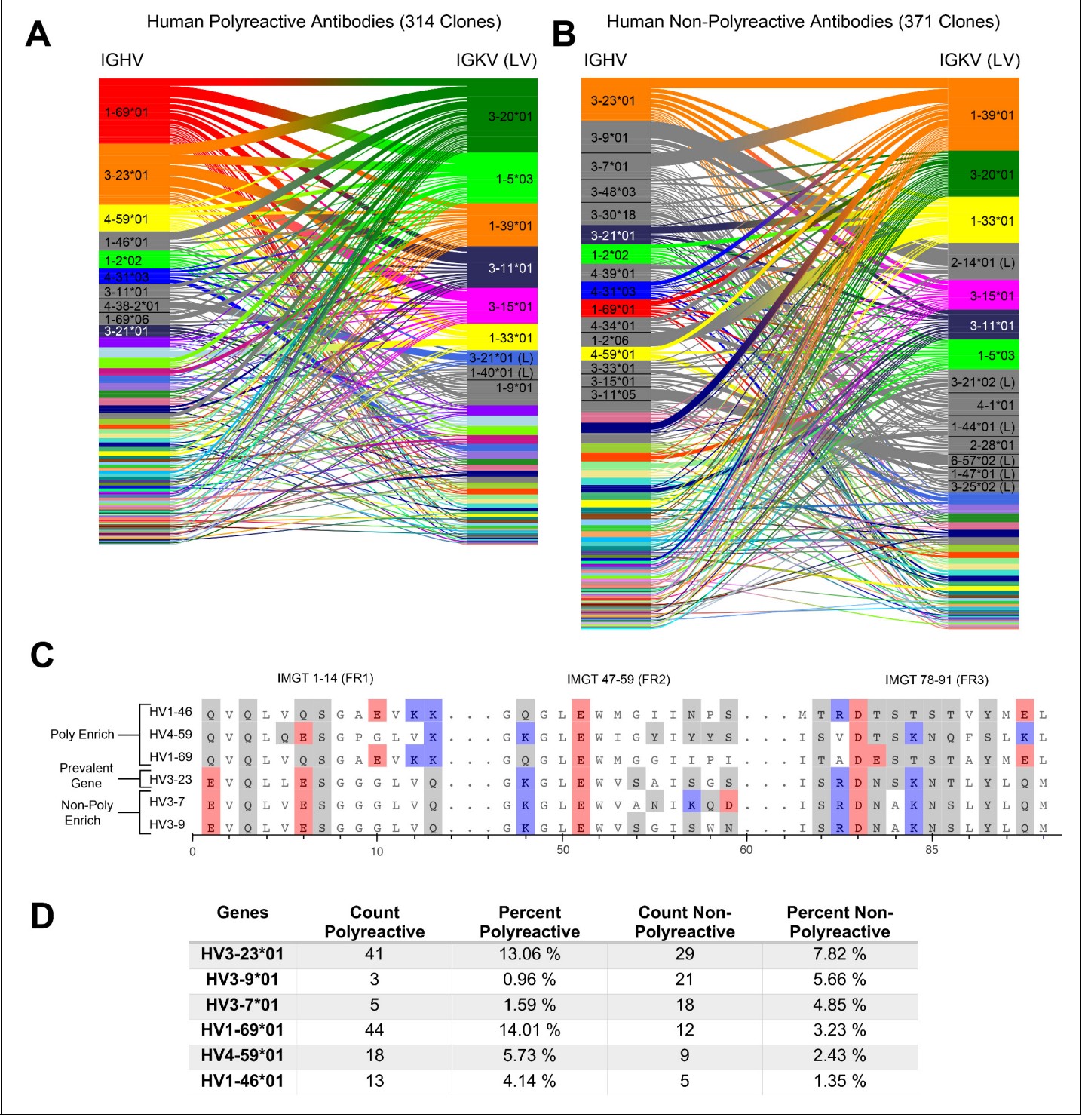

**Figure 1.** A comparative genetic analysis of human-derived polyreactive and non-polyreactive antibody sequences uncovers population level differences. Gene usage diagrams comparing (**A**) human polyreactive and (**B**) non-polyreactive sequences show a qualitative difference in the VH gene usage. Shared colors indicate identical genes, gray indicates genes that are not seen in the other population at a level over 2%. Unlabeled genes are colored randomly to highlight genetic variation in the populations. (**C**) Sequence alignment of the most prevalent genes in the polyreactive and non-polyreactive populations compared to a reference gene common to each population. Hydrophobic amino acids are colored white, hydrophilic amino acids are colored gray, and positively or negatively charged amino acids are colored blue or red, respectively. (**D**) Percentage and raw count of observed gene usage for the polyreactive and non-polyreactive sequences.

The online version of this article includes the following figure supplement(s) for figure 1:

*Figure 1 continued on next page*

*Figure 1 continued*

**Figure supplement 1.** A comparative genetic analysis of the parsed human-derived polyreactive and non-polyreactive antibody sequences uncovers stronger population level differences.

**Figure supplement 2.** A comparative genetic analysis of the mouse-derived polyreactive and non-polyreactive antibody sequences uncovers population differences and a movement away from charged residues in heavy chain poly-enrich genes at IMGT positions 20, 47, 48, 69, and 95.

**Figure supplement 3.** Data from *Figure 1A and B* including J-gene usage, using default TCRdist color scheme.

**Figure supplement 4.** Sequence alignment from *Figure 1C*, including the full amino acid sequences of each heavy chain gene.

**Figure supplement 5.** The raw count of amino acids found in polyreactive and non-polyreactive antibody sequences shows no notable differences.

(472 sequences). Direct comparisons between mouse and human derived antibodies is difficult at the gene usage level. A similar analysis highlighting differences between mouse polyreactive and non-polyreactive antibodies can be found in the supplement (*Figure 1—figure supplement 2*).

Genes are identified from nucleotide sequences using NCBI's IgBLAST command line tool (*Ye et al., 2013*). Heavy and light chain genes that are shared between polyreactive and non-polyreactive sequences are colored for the top labeled instances. Genes that are labeled but not found above a 2% threshold in the opposite population are colored gray, whereas those that do not have a visible name are colored randomly to highlight variation in gene usage. From this comparison, it is clear that the variable gene usage is skewed between polyreactive and non-polyreactive sequences, with an enrichment of $V_H1-69$, $V_H1-46$, and $V_H4-59$ in the polyreactive population, a trend that persists in the parsed dataset (*Figure 1—figure supplement 1*). In contrast, no qualitative differences in the J-gene usage are readily discernible between these two groups (*Figure 1—figure supplement 3*).

While the full alignment of these most used heavy chain variable genes shows a high degree of sequence similarity (*Figure 1—figure supplement 4*), *Figure 1C* highlights the regions of highest dissimilarity between the biophysical properties of amino acids in prevalent genes within each population. $V_H3-23$, the most prevalent gene in the non-polyreactive human dataset and the second most prevalent gene in the polyreactive human dataset, can be used as a reference for comparisons between genes enriched in each individual population. This reference gene shares a high degree of sequence similarity with the second and third most frequently occurring genes in the non-polyreactive dataset, $V_H3-7$ and $V_H3-9$, save for a lysine and aspartic acid pair in framework 2 of $V_H3-7$. The genes enriched in the polyreactive dataset, however, are quite different from this reference. All three of the polyreactive enriched genes have charged residues where the non-polyreactive enriched genes have hydrophilic residues (or vice versa) at IMGT positions 1, 13, and 90. These initial results hint at some systematic differences between the polyreactive and non-polyreactive antibody populations.

*Figure 1D* quantifies the extent of the difference in gene usage in each population by comparing these most prominent genes from our accumulated dataset of HIV- and influenza virus-reactive antibodies. While the two most common genes in the polyreactive dataset account for 27% of the human polyreactive antibodies in this study, the top three most common genes in the non-polyreactive dataset account for just over 17% of the total population. In addition to being the most prevalent gene in the polyreactive dataset, $V_H1-69*01$ has also been found historically to be more prevalent in broadly neutralizing antibodies against influenza viruses, in line with the previously mentioned overlap between bnAbs and polyreactivity (*Andrews et al., 2015*; *Wrammert et al., 2011*).

Overall, there is a noticeable difference between the gene usage frequency of polyreactive and non-polyreactive antibodies, but the overlap in the usage of the two populations suggests that gene usage alone is not sufficient to distinguish the two groups. While there exist qualitative differences between framework sequences enriched in the polyreactive dataset compared to the non-polyreactive population, a look at the amino acid usage of the CDR loops of each group shows no significant differences (*Figure 1—figure supplement 5*). This implies that the positional context of a given amino acid is critical to tease out differences in antibody binding properties.

## A position-sensitive matrix representation of sequences provides further insights into polyreactivity

To identify deeper trends in the biophysical properties of polyreactive antibodies, we utilize a new methodology to analyze and represent a range of different properties inherent to these sequences.

Although the framework regions of antibodies are highly conserved, the CDR loops vary significantly in length and show very low conservation between populations. This makes alignment of CDR loops difficult without creating subgroups for loops of identical length. To overcome this, the sequence data is reorganized into a matrix representation (*Figure 2A*). Each sequence is aligned by the center of each CDR loop, with spaces between the loops set to zero and each amino acid encoded as a number from 1 to 21. While this alignment method excludes the framework regions of the antibodies and slightly averages out some of the properties at the edge of the CDR loops, we reason that most of these differences are evident in the gene usage analysis of the previous section. From this

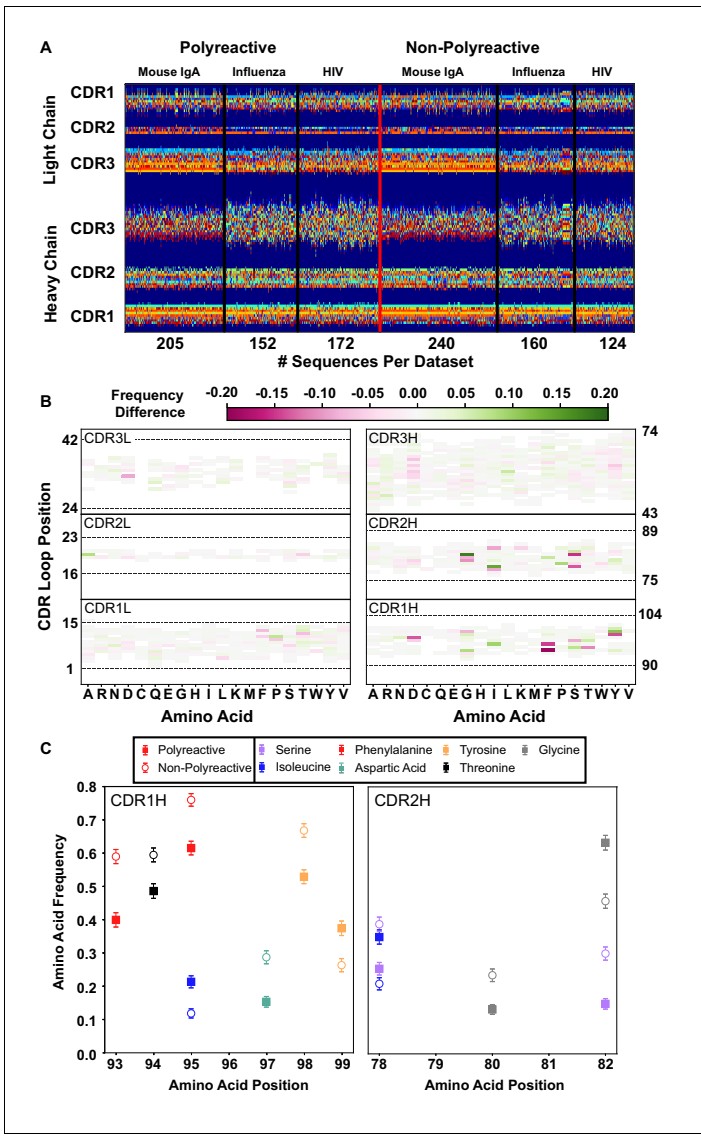

**Figure 2.** A new representation of CDR loop sequences improves the position-sensitivity of quantitative antibody analysis. (A) Matrix representation of the amino acid sequences used in this study provides a framework for further analysis. Each amino acid is encoded as a number from 1 to 21, represented by a distinct color in the matrix. A 0-value is used as a buffer between loops and is represented by the dark blue regions. The red line separates polyreactive and non-polyreactive sequences. (B) Amino acid frequency difference between polyreactive and non-polyreactive sequences for all six CDR loops. Residues more common in polyreactive sequences are shown in green, while those more common in non-polyreactive sequences are shown in pink. Loop positions correspond to the numerical position within the matrix of panel (A). (C) An in-depth representation highlighting the amino acid frequencies used to create panel (B). Only frequency changes greater than 10% are shown for clarity.

The online version of this article includes the following figure supplement(s) for figure 2:

**Figure supplement 1.** Identical analysis to that in *Figure 2* using the parsed dataset displays more pronounced differences between polyreactive and non-polyreactive antibodies.

simple alignment, no obvious patterns emerge separating polyreactive and non-polyreactive antibodies; however, we can clearly see that mouse gut-derived IgA antibodies have generally shorter CDR3H loops, and more conserved CDR3L sequences when compared to the human-derived antibody sequences. All subsequent analysis is derived from this matrix representation of the sequences.

With this new positionally sensitive and quantitative alignment method, we are able to further dissect the differences in amino acid sequences presented in *Figure 1*. *Figure 2B* uses this positional sequence encoding to determine the amino acid frequency difference between polyreactive and non-polyreactive sequences. For example, phenylalanine is found at position 93 in roughly 40% of polyreactive sequences and nearly 60% of non-polyreactive sequences. Therefore position 93, amino acid F has an intensity of −0.2 in *Figure 2B*.

From this panel it is evident that most of the major differences are in the germline encoded regions CDR1H and CDR2H, in line with the observations from *Figure 1* that suggest polyreactive antibodies have a distinct gene usage when compared to non-polyreactive antibodies. *Figure 2C* further expands on these differences, showing the largest changes in amino acid frequencies between the two populations. We can see that there is a slight decrease of phenylalanine frequency in CDR1H of polyreactive antibodies, in favor of isoleucine. Additionally, there is a general shift toward hydrophobicity in CDR2H, as the hydrophilic residue serine at matrix positions 78 and 82 is less prevalent in polyreactive antibodies, instead replaced by the more hydrophobic residues isoleucine and glycine. In the parsed dataset, these differences become larger in magnitude, particularly in CDR1L, where phenylalanine is again found less frequently in polyreactive sequences (*Figure 2—figure supplement 1*).

This increased prevalence in loop hydrophobicity of polyreactive antibodies has been suggested before in the literature (*Prigent et al., 2018*) along with a net increase in positive charge (*Rabia et al., 2018*), so we next aimed to analyze this matrix systematically using biophysical properties inherent to the loops. A simple analysis of the full human and mouse-derived dataset investigating classical parameters explored previously by other groups (CDR loop length, net charge, net hydrophobicity, and gene usage) and some new properties (side chain flexibility, side chain bulk, and the Kidera Factors from *Kidera et al., 1985*) show some significant differences between polyreactive and non-polyreactive antibodies (*Figure 3A,B*). The versatility of the positionally sensitive amino acid matrix allows for the application of multiple 'property masks' to tease out the specific regions of each CDR loop that contributes most to these significant differences. Given a property, amino acid charge for example, we can replace each simple 1–21 representation with a distinct representation based upon amino acid properties.

In the matrix of *Figure 2A* leucine, histidine, and arginine are represented by the integers 3, 16, and 17. As an example, when the charge property mask is applied, the matrix representations of these three amino acids in all sequences is changed to 0.00, 0.091, and 1.00, respectively. We apply 62 such masks to this matrix, including simple metrics like charge, hydrophobicity, side chain flexibility, and side chain bulkiness to go along with more carefully curated metrics from the works of *Kidera et al., 1985*; *Liu et al., 2018*. A complete description of these properties can be found in the 'Key resources table' and in *Appendix 1—table 1*. The application of these masks gives an entirely new matrix describing the localization of amino acids with a given property.

By averaging across all sequences in the polyreactive or non-polyreactive dataset when these masks are applied, we can readily see differences in charge patterning and hydrophobicity when comparing polyreactive and non-polyreactive sequences (*Figure 3C,D*).

Including errors obtained via bootstrapping, we see that these differences are most pronounced in the center of CDR3H, with some differences also apparent in the remaining five loops. This analysis shows an overall bias toward neutrality in these regions; that is neither positively nor negatively charged, neither strongly hydrophilic nor hydrophobic. These results also contextualize the findings of *Figure 2C*. The trend toward hydrophobic residues in CDR2H of polyreactive antibodies importantly does not make these regions net hydrophobic, but instead make these regions slightly less hydrophilic on average. This effect is yet again more pronounced in the parsed dataset (*Figure 3—figure supplement 1*), with a strong trend toward interface neutrality. Conversely, when comparing bootstrap samples drawn from the null distribution, that is the 'polyreactive' or 'non-polyreactive' labels are given to antibody sequences at random, we see no difference between the biophysical properties of the two populations (*Figure 3—figure supplement 2*).

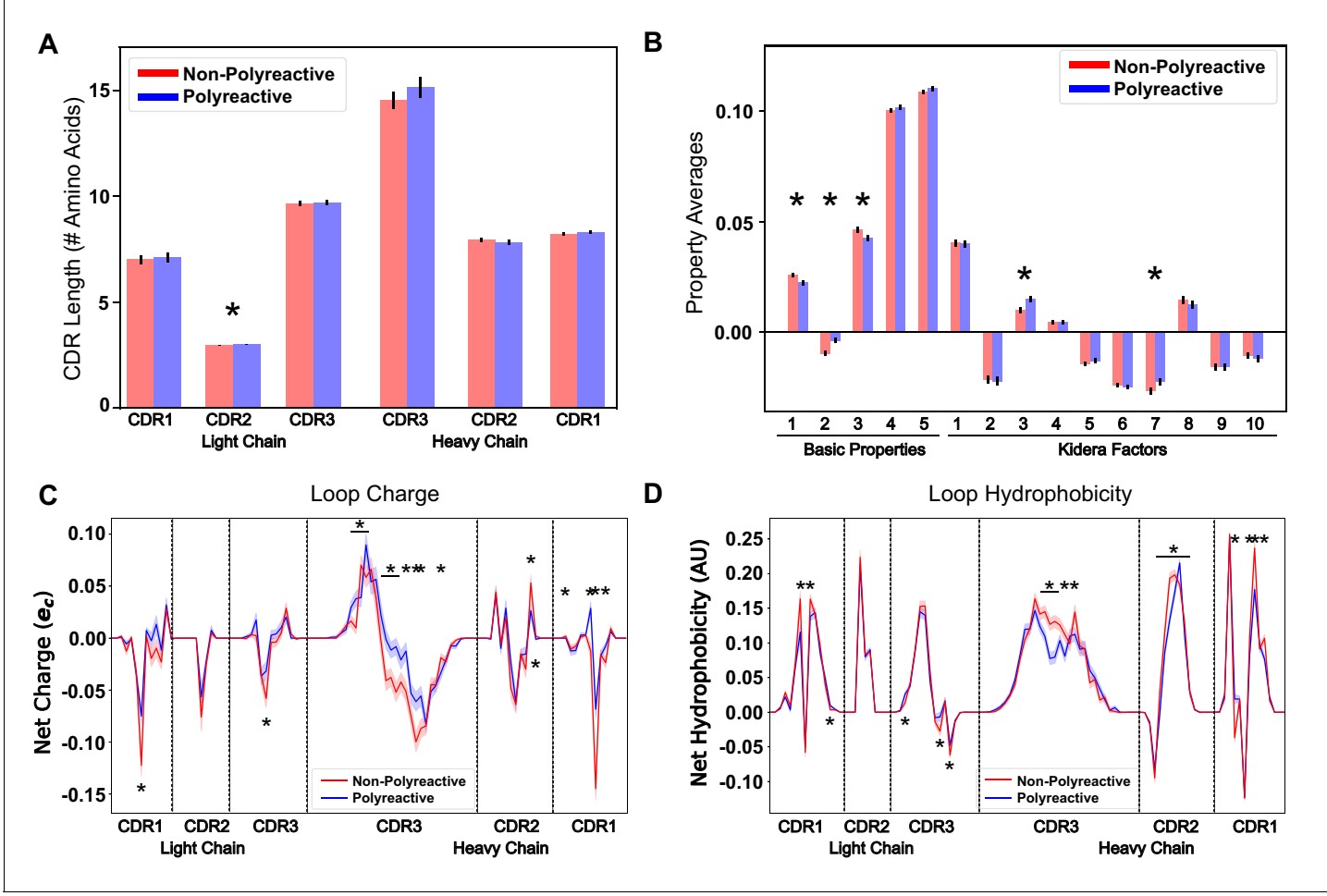

**Figure 3.** Position-sensitive quantification of CDR loop properties of mouse and human antibody sequences highlights differences between polyreactive and non-polyreactive populations. Plotting the average CDR loop lengths (A) and net antibody biophysical properties (B) show small but significant differences when analyzed in bulk. Basic properties 1–5 are hydrophobicity1, charge, hydrophobicity2, side chain flexibility, and side chain bulk. Plotting the average net charge (C) and hydrophobicity (D) as a function of position of polyreactive and non-polyreactive sequences highlights significant differences in CDR3H. Light shadow around lines represent bootstrap standard errors. All uncertainties obtained via bootstrapping. Stars indicate p-value ≤ 0.05 calculated via nonparametric Studentized bootstrap test. Bars with a single star above represent contiguous regions of significance. p-values in panels (A) and (B) corrected for multiple tests using the Bonferroni correction.

The online version of this article includes the following figure supplement(s) for figure 3:

**Figure supplement 1.** Identical analysis to that in *Figure 3* using the parsed dataset displays more pronounced differences between polyreactive and non-polyreactive antibodies.

**Figure supplement 2.** Identical analysis to that in *Figure 3* using bootstrapped means drawn from the null distribution.

## Systematic determination of the key contributions to polyreactivity

Along with simple property averaging, these masks also give a high dimensional space from which we can determine, in an unbiased way, the primary factors that discriminate polyreactive and non-polyreactive antibodies. As a first pass, we apply a principal component analysis (PCA) to the matrix of all antibody sequences in an attempt to separate the polyreactive or non-polyreactive populations along the axes of highest variation in the dataset. Unfortunately, the principal components of these data do not effectively distinguish between the two populations (*Figure 4—figure supplement 1*).

To further investigate the physical and sequence-based properties of polyreactivity in antibodies in a more targeted manner, we employ linear discriminant analysis (LDA), a common algorithm often applied in classification problems (*Barker and Rayens, 2003*; *Cordeiro et al., 2009*; *Ma et al., 2013*). LDA works in a manner conceptually similar to PCA, reducing the dimensionality of a given dataset via a linear combination of the original dimensions. However, LDA takes one additional

input, the label or class of each sequence. Whereas the objective of PCA is to identify the axes which maximize the variance in the dataset, LDA has the dual objective of maximizing the projected distance between two classes while minimizing the variance within a given class. While LDA is well adapted for classifying two distinct populations, it is susceptible to overfitting, unlike PCA (*Qiao et al., 2009*). Here, we have labeled our two classes in the matrix with either a '1' for polyreactive, or '0' for non-polyreactive. In our application of LDA, we parse down the large number of input vectors using either PCA or an algorithm which selects the vectors with the largest average differences between the two populations. This reduction in dimensionality ensures the data are not being overfit, and the tunable number of input vectors allows us to control for overfitting in each individual application.

*Figure 4A* shows the results of LDA when applied to the parsed dataset comprised of 311 polyreactive antibodies and 362 non-polyreactive antibodies. As discussed in the introduction, the framing of polyreactivity as a binary problem is not a perfect assumption. The inclusion of intermediate levels of polyreactivity further confounds this issue. Indeed, the application of LDA to the full dataset shows a reduced ability to split polyreactive and non-polyreactive antibodies (*Figure 4—figure supplement 2*), likely due to this spectrum of polyreactivity. By considering only the parsed dataset for these classification analyses, we can improve confidence that the differences identified are those that separate strongly polyreactive and strongly non-polyreactive antibodies.

LDA analysis is versatile in its applications, and in this work, we utilize the method in two distinct modes. In the first mode, all available data is used as input with the output vector representing the features that best distinguish between the two complete populations. Plots of the data projected onto this vector (as in *Figure 4A*) represent the maximum achievable separation between the two populations for a defined number of input components from the given biophysical property matrix. In the second mode, we utilize LDA as a more canonical classification algorithm separating the data randomly into training and test groups. In this classification mode of operation, a combination of correlation analysis coupled with maximal average differences is used to parse input features, and a support vector machine (SVM) is used to generate the final classifier from these features. Accuracy of the resultant classifiers is assessed via leave one out cross validation, these accuracies are shown in *Figure 4B*.

In the first mode, we find that the data can be split more effectively when the parsed dataset is broken up into the distinct 'reactivity' groups, that is those antibodies specific for influenza viruses, HIV, or found in the mouse gut (*Figure 4A*). This suggests there may be some bias due to antigen specificity, or lack thereof, whereby influenza virus-specific antibodies take a slightly different path toward polyreactivity compared to HIV reactive or mouse gut IgA antibodies. However, when using the classification mode, the classification accuracy is roughly equivalent across all tested datasets (*Figure 4B*). Testing this classifier with a scrambled dataset, where the labels are randomly assigned, shows the expected decrease in classification accuracy for each individual dataset for all ranges of input features.

When applying LDA in the first mode (*Figure 4A*), we can directly pull the linear weights of each component comprising linear discriminant one and reveal which biophysical properties at each CDR position best distinguish between the two populations. The differences in the linear weights from the heavy chain CDR loops comprising each discriminant show clear differences when comparing the complete parsed dataset (*Figure 4C*) to the HIV only dataset (*Figure 4D*). In the parsed dataset, the discriminating weights are heavily concentrated in CDR2H. Whereas in the HIV dataset, these weights are centered around the CDR3H loop. Only the top 10 linear weights are shown in *Figure 4C,D*. The full matrix of linear weights can be found in *Figure 4—figure supplement 3*. The predominant discriminating factors between datasets might be due to the significant difference in CDR3H length between the mouse (IgA) and the human datasets, which confounds the analysis in this region. However, when examining each individual subset of the complete dataset we do find that there are common properties that seem to be the primary discriminators (i.e. largest linear weights). These are hydrophobicity 1, hydrophobicity 2, and hotspot variable 6 (a structural parameter related to α-helix propensity).

## An information theoretic approach

While analysis of the biophysical property differences between polyreactive and non-polyreactive sequences provides some insight into the molecular basis for the polyreactivity phenomenon, a

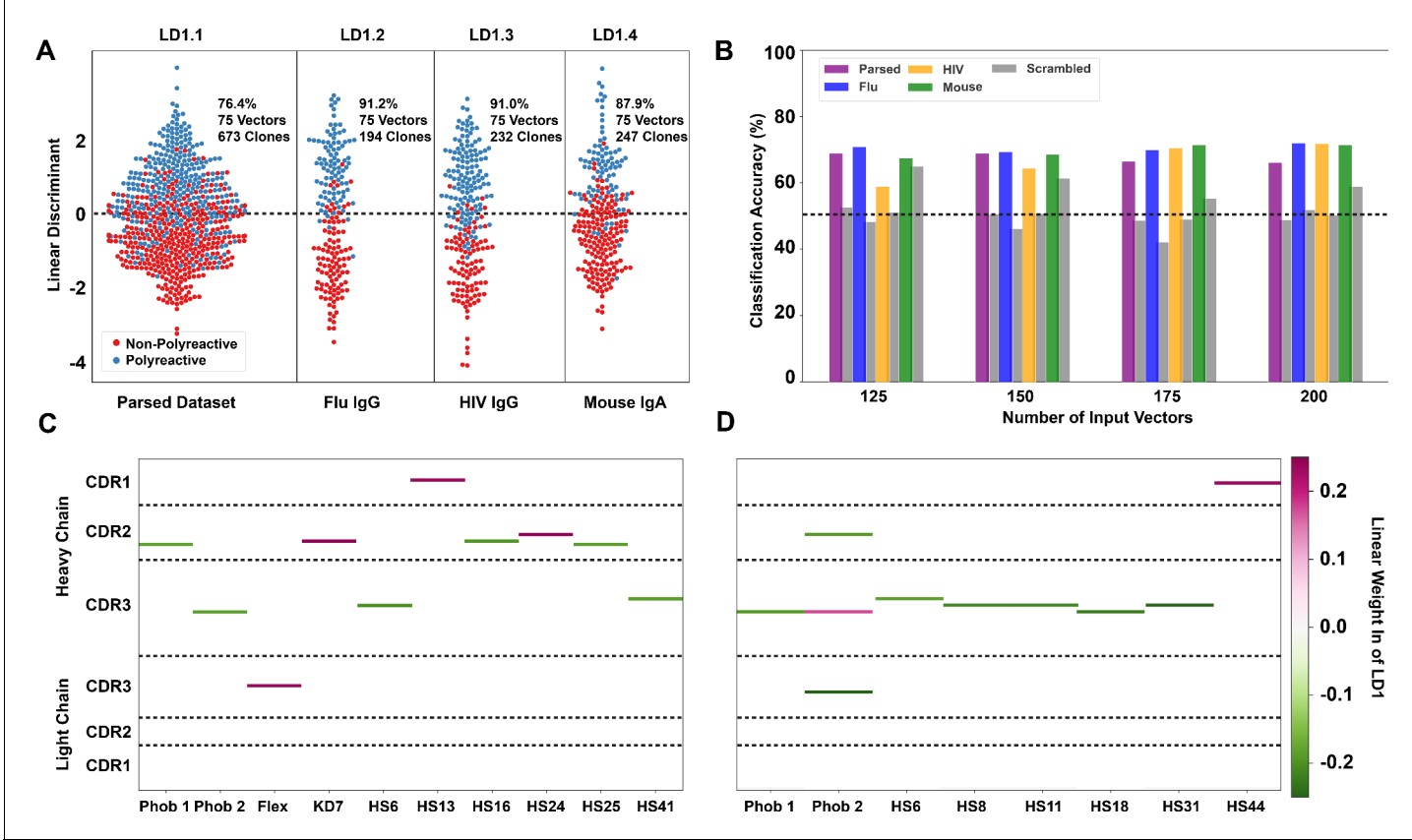

**Figure 4.** Linear discriminant analysis (LDA) can meaningfully separate the two populations and these meaningful differences can be used to generate a polyreactivity classifier. (**A**) LDA applied individually to the complete parsed, Influenza, HIV, and mouse datasets. Percentages indicate the accuracy of the linear discriminant in labeling polyreactive and non-polyreactive antibodies. For these data, the plotted linear discriminants are comprised of different linear weights. (**B**) Accuracies of a polyreactivity classifier with a separate test and training dataset. Groupings in this figure are the same as those in panel (**A**). A support vector machine is generated for each individual population, and the reported values are accuracies calculated through leave one out cross validation. Shown are test data and a scrambled dataset where the labels of 'polyreactive' or 'non-polyreactive' are applied randomly (gray bars). The dotted line indicates 50% accuracy threshold. (**C**) Property matrices highlighting the top 10 weights of the linear discriminants in panel A for the parsed dataset with 75 vectors (**C**) and the HIV dataset with 75 vectors (**D**). Color bar represents the normalized weight of each property, where pink rectangles represent properties correlated with increased polyreactivity, and green rectangles represent properties correlated with decreased polyreactivity.

The online version of this article includes the following figure supplement(s) for figure 4:

**Figure supplement 1.** Principal component analysis (PCA) applied to the full amino acid usage matrix and the top 75 discriminating vectors used for linear discriminant analysis.

**Figure supplement 2.** Analysis using the same approach as *Figure 4* applied to the complete dataset of 1053 polyreactive and non-polyreactive antibodies highlights the expected decrease in classification accuracy when considering intermediate levels of polyreactivity.

**Figure supplement 3.** The complete representation of the 75 linear weights that most effectively separate polyreactive and non-polyreactive sequences in the parsed complete dataset (**A**) and the parsed HIV dataset (**B**).

broad unifying pattern which could discern the biophysical mechanism behind polyreactivity was not readily evident across all types of antibodies. To probe these polyreactive sequences in a quantitative yet more coarse manner, we applied the formalism of information theory to our dataset of antibody sequences. Information theory, a theory classically applied to communication across noisy channels, is incredibly versatile in its applications, with high potential for further applications in immunology (*Shannon, 1948*; *Román-Roldán et al., 1996*; *Cheong et al., 2011*; *Vinga, 2014*; *Mora et al., 2010*; *Murugan et al., 2012*). In this work, we utilize two powerful concepts from information theory, namely Shannon entropy and mutual information.

Shannon entropy, in its simplest form, can be used as a proxy for the diversity in a given input population. This entropy, denoted as H, has the general form shown in Equation 1:

$$H(X) = -\sum_X p(x) \log_2 p(x) \tag{1}$$

where $p(x)$ is the occurrence probability of a given event, and $X$ is the set of all events. We can then calculate this entropy at every position along the CDR loops, where $X$ is the set of all amino acids, and $p(x)$ is the probability of seeing a specific amino acid at the given position. In other words, we want to determine, for a given site in a CDR loop, how much diversity (or entropy) is present. *Figure 5A* shows this Shannon entropy distribution for the full dataset of polyreactive and non-polyreactive antibodies. Given there are only 20 amino acids used in naturally derived antibodies, we can calculate a theoretical maximum entropy of 4.2 bits, which assumes that every amino acid occurs at a given position with equal probability. Although the observed entropy of the CDR3H loop approaches this theoretical maximum, it hovers below it (3.5 Bits) due to the relative absence of the amino acids cysteine and proline in the center of this loop. The difference in the entropy distributions in CDR1H are consistent with the bias in amino acid usage in this region, shown previously in *Figure 2*.

Importantly, from this entropy we can calculate an equally interesting property of the dataset, namely the mutual information. Mutual information is similar, but not identical to, correlation. Whereas correlations are required to be linear, if two amino acids vary in any linked way, this will be reflected as an increase in mutual information. In addition, due to some of the highly conserved residues in the non-CDR3H loops, high covariance can be achieved for residues that have not been specifically selected for in the germinal center. Using this information theory framework, these conserved residues have a mutual information of 0. Overall, the mutual information can be used to identify patterns in antibody sequences that were not readily evident through the previous analysis in this or other studies. If there is some coevolution or crosstalk between residues undergoing some selection pressure in the antibody maturation process, it will be reflected as an increase in the mutual information. In this work, mutual information $I(X;Y)$ is calculated by subtracting the Shannon entropy described above by the conditional Shannon entropy $H(X|Y)$ at each given position as seen in *Equations 2 and 3*:

$$H(X|Y) = -\sum_{y \in Y} p(y) \sum_{x \in X} p(x|y) \log_2 p(x|y) \tag{2}$$

$$I(X;Y) = H(X) - H(X|Y) \tag{3}$$

To orient ourselves in physical space, *Figure 5B* gives an example crystal structure (PDB: 5UGY) (*Whittle et al., 2011*; *Ziegler et al., 2014*) highlighting the lateral arrangements of the CDR loops. The matrix in *Figure 5C* shows that the mutual information between CDR loops on this binding surface is increased in the heavy chains of polyreactive antibodies over non-polyreactive ones, suggesting an increase in loop crosstalk in antibodies that exhibit polyreactivity. Interestingly, it appears that there is a corresponding decrease of loop crosstalk in the light chains of polyreactive antibodies. Importantly, this crosstalk is increased across and within all loops when analyzing the parsed dataset (*Figure 5—figure supplement 1*). This observed crosstalk persists across all polyreactive antibodies within all subsets of our tested dataset and is evident both in intra-loop and inter-loop interactions. *Figure 5D* highlights some examples of the interesting significant differences of this crosstalk at distinct given positions within CDR1L, CDR1H, and CDR3H. A complete plot of the statistically significant differences ($p \leq 0.05$) of *Figure 5C* shows that a large portion of these differences are in fact significant (*Figure 5—figure supplement 2*).

The ordering of these entropy and information plots was chosen to reflect the spatial arrangement of the loops on the antibody surface; as such they show also that mutual information between loops drops off with physical distance between these loops. In other words, loops (and residues) that are located close to each other will have more of an effect on their direct neighbors as opposed to those that are more physically distant. This increased mutual information suggests that in the heavy chains of polyreactive antibodies, there is enhanced cooperativity or co-evolution of the amino acids of intra- and inter-CDR loop pairs.

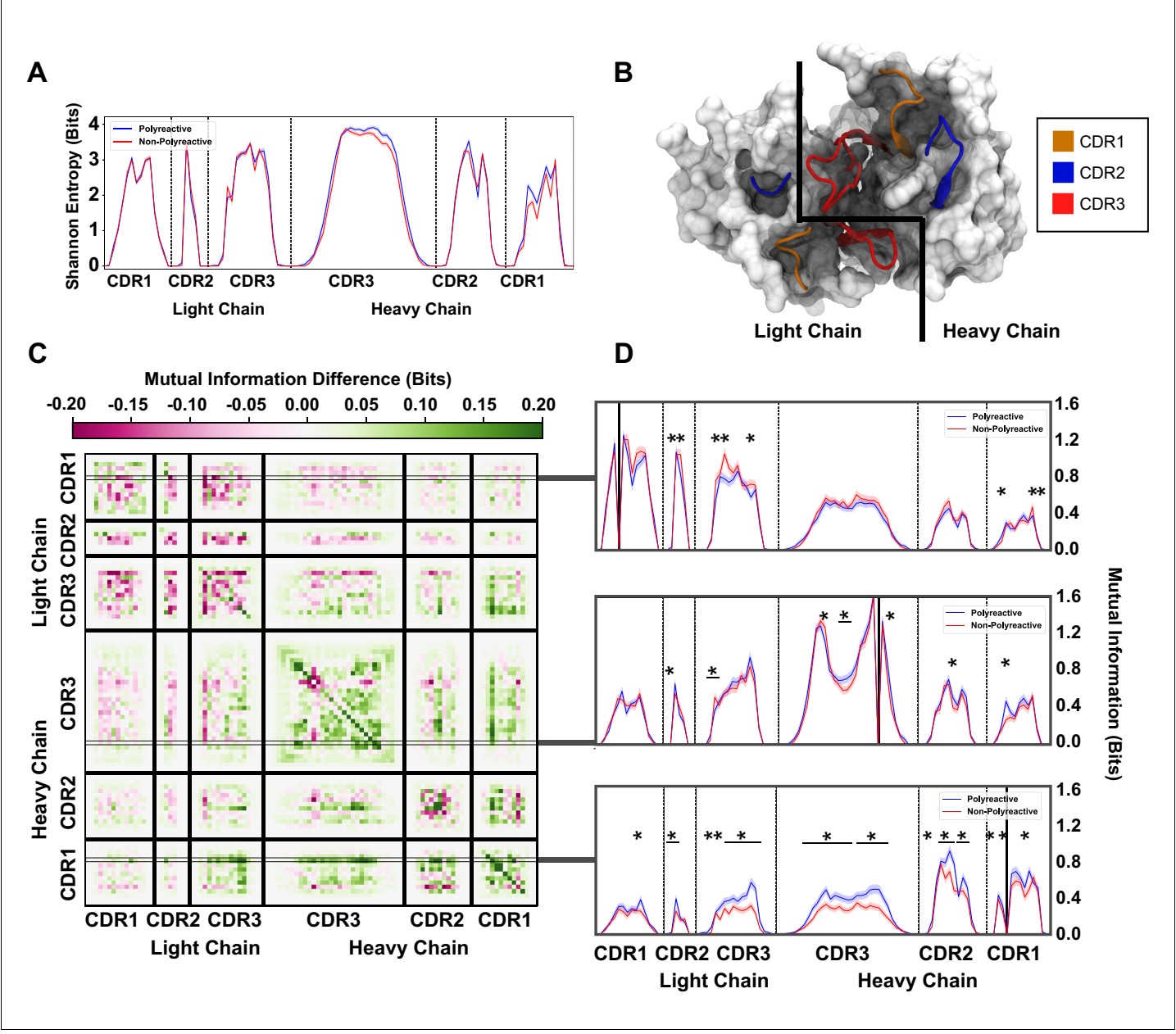

**Figure 5.** An information theoretic analysis of antibody sequences shows an increase in polyreactive antibody loop crosstalk. (A) The sequence diversity of the polyreactive and non-polyreactive datasets, quantified using Shannon Entropy, highlight similar diversities between the two groups. (B) A crystal structure (PDB: 5UGY) provides a visual representation of the lateral organization of the CDR loops on the antibody binding surface. (C) The difference in mutual information between polyreactive and non-polyreactive sequences shows that CDR loops of the heavy chain have more crosstalk in polyreactive antibodies. Each individual row represents the given condition, whereas each column gives the location the mutual information is calculated. (D) Singular slices of the mutual information show the data in (C), projected from the matrix onto a line, highlighting the significance of the differences at these particular locations. The positions of the 'given' amino acid, that is the particular $Y$ in $H(X|Y)$, are highlighted by gray boxes in panel C. Solid black lines indicate where on the X-axis this 'given' amino acid is located. Stars indicate statistical significance ($p \leq 0.05$) calculated through a nonparametric permutation test. Bars with a single star above represent contiguous regions of significance.

The online version of this article includes the following figure supplement(s) for figure 5:

**Figure supplement 1.** An information theoretic analysis of the parsed antibody sequences shows an increase in polyreactive antibody loop crosstalk that is more pronounced when compared to the full dataset.

**Figure supplement 2.** The statistical significance of the values reported in *Figure 5C*.

## Extension of the analysis to MHC and MHC-like molecules

Given the ability of this analysis pipeline to find nuanced differences between polyreactive and non-polyreactive antibodies, we next sought to expand the range of applications of our approach. Extension of the pipeline to the analysis of TCR sequences is trivial, due to the similar arrangement of CDR loops on the binding surface and the capability of IgBLAST to annotate TCR sequences (*Ye et al., 2013*). Instead, we sought to significantly expand the scope of this software by applying a similar approach to the analysis of MHC and MHC-like molecules. MHC molecules are encoded by a large superfamily of genes that are spread throughout the genome (*Adams and Parham, 2001*; *Piertney and Oliver, 2006*). MHC and MHC-like genes are found across a wide range of divergent species, and these genes have diversified extensively over time, making the distinction between orthologs and instances of convergent evolution difficult. In some cases, the divergence is extreme enough that phylogenetics cannot provide predictions of function. Given that these MHC molecules have evolved to present different antigen subtypes, such as lipid molecules in the case of CD1 proteins (*Borg et al., 2007*; *Luoma et al., 2013*; *Adams, 2014*), we explored the use of our pipeline as a classifier based on biophysical properties rather than phylogeny. In achieving this new functionality, the critical step lies in the transformation of the MHC sequences into a numeric form as in *Figure 2A*.

To accomplish this, we split the sequences by their most prominent structural features. For MHC and MHC-like molecules, these features are the two β-strands and α-helices of the so-called platform domain. As a test case, we use two example training classes; a representative list of human MHC Class I molecules, and the output from a BLAST query on CD1d (*Sayers et al., 2020*). We can then assess our ability to separate these two training classes, while also introducing a test class derived from the data of Almeida et al. (*Almeida et al., 2020*). Each sequence within a given class is globally aligned, and one representative sequence from each class is sent through the Phyre2 structural prediction server (*Kelley et al., 2015*). We then use these structural predictions to identify the start and end points of each major structural feature in the alignment space. These start and end points then define the boundaries that are numerically encoded into our position-sensitive matrix, as seen in *Figure 6A*.

Once the data are in this form, all downstream analysis outlined previously can be applied in a similar manner. In this example case, we find that average biophysical properties across these sequences reveal expected differences in hydrophobicity, with the lipid binding CD1 molecules displaying increased hydrophobicity when compared to the peptide binding MHC class I molecules (*Figure 6B*). Interestingly, unlike in the case of the antibody analysis, a simple PCA can effectively discriminate between the two training classes in this example case. *Figure 6C* shows the projection of the biophysical property data of each class onto the first two principal components. Here, we see that the peptide binding molecules (HLA-E, HLA-A2, H2-D) and the lipid binding molecules (Human CD1b, Chicken CD1) of the test dataset cluster with the respective peptide and lipid binding training data. The majority of the data of Almeida et al., comprised of non-classical MHC class I molecules from cartilaginous fish, clusters as its own distinct group, likely due to evolutionary distance between these molecules and those derived from mammalian and avian immune systems.

## Discussion

Previous research has highlighted the importance of hydrophobicity, charge, and CDR loop flexibility on antibody specificity. In this work, we expand upon these previous results with a new bioinformatic and biophysical characterization of polyreactive antibodies. The software generated for this study provides a powerful computational tool which can be utilized by researchers interested in discerning differences between populations of adaptive immune molecules in broad contexts. Building off of the efforts of our own work and that of experimental collaborators, we were able to aggregate to date one of the largest publicly available datasets of antibodies tested for polyreactivity. Differences in the germline gene frequency and amino acid frequencies show there exists some underlying differences between polyreactive and non-polyreactive antibodies. A surface level analysis of this dataset is able to discriminate certain features of polyreactive and non-polyreactive antibodies, namely that on average, polyreactive antibodies are less strongly negatively charged, less hydrophilic, and have a higher prevalence of antibodies with longer CDR loops of the heavy chain. Importantly, however, these binding surfaces do not have a net positive charge nor are they net hydrophobic.

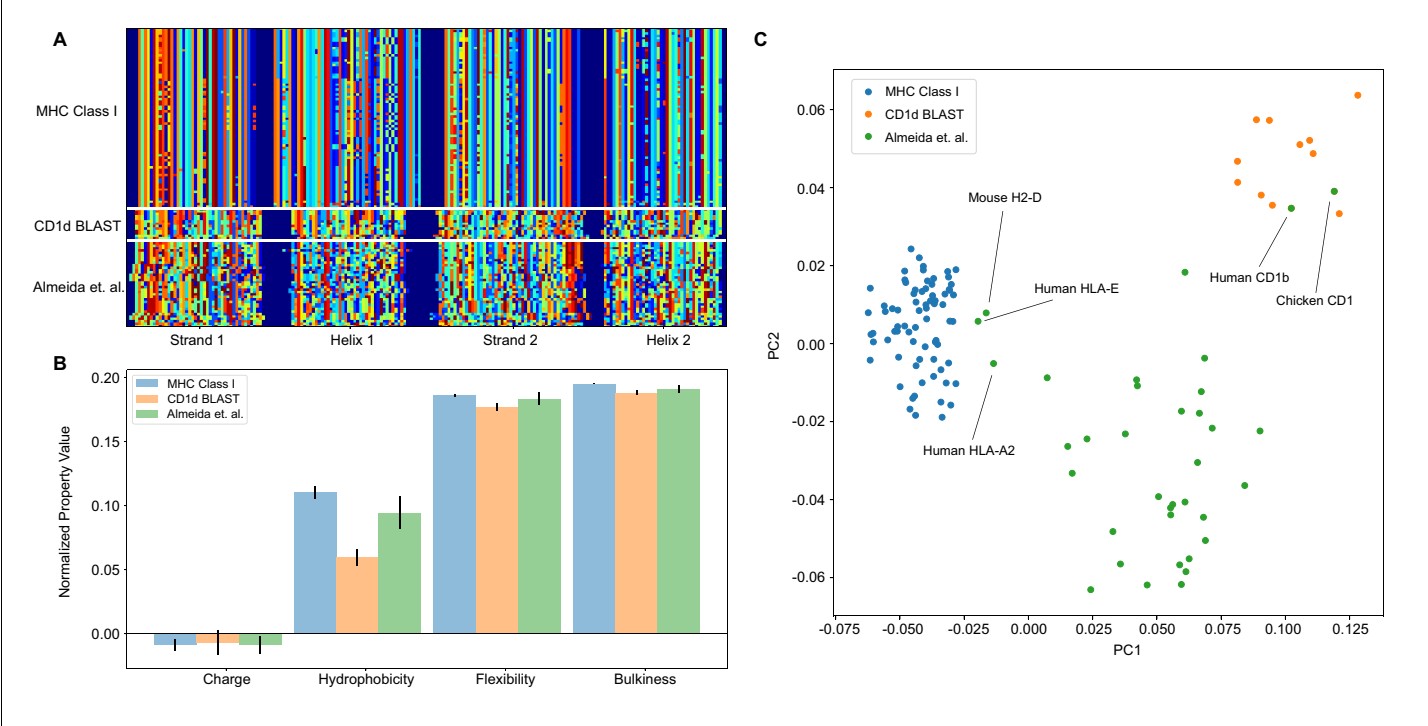

**Figure 6.** The analysis pipeline is flexible and sufficiently identifies differences between MHC Class I and CD1-like molecules, and has the potential to be used as a classifier moving forward. (A) Similar to antibody sequences, the MHC and CD1 sequences can be encoded into a matrix. Here, we focus on human Class I MHC molecules and CD1 molecules from various organisms, and use these sequences as training data to classify sequences collected in *Almeida et al., 2020*.(B) Comparisons of simple biophysical properties across these molecular species highlight differences between classes. (C) Projection of the biophysical properties of each class on to the first two principal components can be used to classify MHC- and CD1-like molecules. Molecules present in *Almeida et al., 2020* but absent from the training data are labeled.

Our results highlight an increase in $V_H1-69$ gene usage in polyreactive antibodies, an interesting finding given the substantial literature outlining its importance in diverse immune environments. In addition to the aforementioned role of $V_H1-69$ in broadly neutralizing anti-influenza and anti-HIV antibodies (*Haynes et al., 2005*; *Mouquet et al., 2011*; *Andrews et al., 2015*; *Prigent et al., 2018*), autoreactive chronic lymphocytic leukemic B cells commonly express receptors bearing $V_H1-69$ (*Sasso et al., 1993*; *Forconi et al., 2010*), and anti-HIV antibodies which target the membrane-proximal external region of HIV-1 envelope glycoproteins frequently utilize $V_H1-69$ (*IAVI Protocol G Investigators et al., 2019*). While previous reports suggest that the key feature permitting these auto-reactive or polyreactive interactions of $V_H1-69$ is an exceptionally hydrophobic CDR2H loop (*Chen et al., 2019*) our results suggest this does not explain the over-representation of this antibody in the polyreactive dataset, as on average the CDR2H of polyreactive antibodies is strongly hydrophilic. Instead, certain structural or dynamic features of the antibody may contribute to its out-sized role in critical biological contexts.

To dig deeper into the biophysical differences between polyreactive and non-polyreactive antibodies, we created an adaptable software for the automated analysis of large antibody datasets and the application of a new analysis pipeline for the study of polyreactive antibodies. Overall, the improvements of this software to the current state of antibody sequence analysis are sufficient to highlight key differences in the two populations with improved spatial resolution. The position sensitive sequence alignment is able to further parse through the genetic differences and show that in general, polyreactive antibodies have a tendency to have more hydrophobic residues in CDR2H, and a decreased preference for phenylalanine in CDR1H. While these observational differences provided some initial insight, a more rigorous biophysical treatment was necessary. With the addition of 62 biophysical properties analyzed using the position sensitive alignment, significant differences between the CDR3H loops in polyreactive and non-polyreactive antibodies become immediately evident, providing a more detailed depiction of the antigen binding surface of polyreactive antibodies.

These data suggest a movement toward neutrality or 'inoffensive' residues in the CDR loops of polyreactive antibodies: amino acids that are neither exceptionally hydrophobic nor hydrophilic and with a net charge close to 0. Previous studies have suggested that polyreactive antibodies tend to have more hydrophobic CDR loops, such that low-affinity Van der Waals interactions might be the primary means of polyreactive interactions (*Prigent et al., 2018*; *Starr and Tessier, 2019*). However, these studies counted the number of hydrophobic residues per sequence or averaged the hydrophobicity of all six CDR loops. While our results partially agree with these previous findings, our analysis extends much further into defining the biophysical basis of this phenomenon. For example, while our position sensitive representation of the sequences shows that CDR3H does become more hydrophobic in polyreactive sequences, it is still net hydrophilic on average. A highly hydrophobic binding surface would provide an avenue for non-specific interactions with other hydrophobic proteins, but it would occlude binding to highly hydrophilic ligands like DNA. A slightly hydrophilic, neutral-charged binding surface would permit weak interactions with a wide range of ligands.

Using these and other biophysical properties as input feature vectors, we were able to generate a generalizable protocol for binary comparisons between two distinct populations of Ig-domain sequences. This framework is able to successfully split all tested polyreactive and non-polyreactive antibody datasets. Care was taken to not overfit these data and a preliminary classifier built from this algorithm was able to identify the proper number of input vectors for each LDA application. While there are general features which best split the polyreactive and non-polyreactive antibodies in these datasets, including charge, hydrophobicity, and α-helix propensity, these features alone are not sufficient to discriminate between the two populations. Instead, 75 vectors taken from the position-sensitive biophysical property matrix are necessary to properly split the groups, including both simple properties like charge, hydrophobicity, flexibility, and bulkiness and more carefully curated properties like the often used Kidera factors and the hotspot detecting variables of Liu et al. (*Kidera et al., 1985*; *Liu et al., 2018*; *Vihinen et al., 1994*). The inability to arrive at a core few biophysical properties that could effectively distinguish polyreactive and non-polyreactive antibodies necessitated the application of further approaches, namely information theory.

The tools provided by information theory proved to be effective in the present study. The classic approach to information theory considers some input, communication of this input across a noisy channel, and then reception of a meaningful message from the resultant output. We can think of the analogous case for these antibodies, whereby the sequence and structure of the antibodies can be seen as our input, the thermal noise inherent to biological systems can complicate biochemical interactions, and the necessary output is antigen recognition, i.e. binding between the antibody and the ligand. Focusing just on the antibody side of this communication channel, we determined the underlying loop diversity through the Shannon entropy of the polyreactive and non-polyreactive datasets. This diversity was found to be nearly equivalent while the mutual information, a metric of 'crosstalk' across populations, between and within CDR loops was found to be increased in the heavy chain and decreased in the light chain of polyreactive antibodies. What this loop crosstalk entails physically is not immediately clear from these measurements.

The mutual information increase could come from gene usage being somehow coupled, amino acid usage coupling with the cognate ligand, or the amino acids directly interacting physically with each other. In some way, this crosstalk appears to be selected for in the polyreactive population. If this increase in mutual information manifests as an increase of charge-charge interactions, this could explain why there is a minimal change in net charge of antibodies between the two groups, yet a significant move toward neutrality in the CDR loops of polyreactive antibodies. The pairing of two charged groups would help move the binding surface of polyreactive antibodies toward a more 'inoffensive' binding surface. A binding surface that is neither exceptionally hydrophobic nor hydrophilic, and lacks a significant positive or negative charge, would represent a relatively appealing binding interface for a low-affinity interaction with a large array of diverse ligands. A patchwork of hydrophobic and hydrophilic non-charged residues exposed to potential ligands would provide an ideal candidate polyreactive surface. The corresponding decrease in the mutual information between the light chain CDR loops of polyreactive antibodies could be caused by a de-emphasis in the involvement of these loops due to differential binding configurations of polyreactive ligands, as has been previously hypothesized (*Dimitrov et al., 2013*; *Sethi et al., 2006*).

While further crystallographic, biochemical, and dynamic studies are necessary to identify the true source of this increase in mutual information across polyreactive antibodies, we can speculate what

these results may mean in the context of the results obtained using linear discriminant analysis. In addition to standard side chain properties, many of the most important features for splitting polyreactive and non-polyreactive antibodies were structural in nature. Specifically, hotspot variables 6, 24, 25, and 41 all correspond to the structural tendencies of a given amino acid. Coupled with the increase in side chain interactions that may be implied by the increased mutual information across the loops of polyreactive antibodies, this potential for increased loop structure may suggest more rigid CDR loops in polyreactive antibodies.

In addition to the insights into polyreactivity, the computational tools developed for this study are broadly applicable to future studies of large antibody or T cell receptor repertoires. One of the strengths of this approach is a decreased emphasis on structural information when crystal structures are unavailable. Computational prediction of loop conformation is difficult, and drawing inferences from incorrect models regarding side-chain interactions and positioning could be misleading. Reliable structural information on these polyreactive antibodies will be critical to a further understanding of the mechanisms of polyreactivity, including complex structures of antibodies bound to various ligands. In the high-throughput analysis of antibody sequences, our approach strikes a careful balance of the structural assumptions that should apply consistently across antibody populations.

Further experimental assays will be necessary to more comprehensively identify the underlying mechanisms of polyreactivity, including further sequencing and biochemical analysis of polyreactive and non-polyreactive antibodies. Antibodies specific to other pathogens or those from other organisms tested for polyreactivity will help form a more complete picture and improve the generality of the results. As with any machine learning based approach, the classification algorithm is only as good as the data it is trained on. Adding further data in the training set, including more mutations and germline reversions that turn a polyreactive antibody non-polyreactive or vice-versa, will be critical for a comprehensive analysis of polyreactivity. Additionally, a more robust assay for determining polyreactivity such as a chip based screen to test for binding to many diverse targets, would greatly broaden our perspective and help understand just how broadly reactive these polyreactive antibodies are. Lastly, a more complete understanding of the germinal center and the selection processes inherent to the affinity maturation process will assist in the determination of whether polyreactivity is a byproduct or a purposeful feature of the affinity maturation process.

The software generated for this study is publicly available as a python application (see Materials and methods). The unique aspect of this software is its hybrid approach to position-sensitive amino acid sequence analysis. Structural information is implicitly encoded by the alignment strategy employed, yet these assumptions are weaker than those imposed by explicit structural prediction. Downstream analysis from this positional encoder is streamlined and can be generalized to analyze any binary or higher order classification problems. This streamlined analysis allows for the generation of each figure in this study to be applied to thousands of sequences in a matter of minutes. The classification capabilities of the software could prove particularly useful when comparing binary classes, such as T cell receptors or antibody sequences derived from healthy and diseased tissue samples. Acceptable inputs are not restricted to CDR loops of immunoglobulins, and we have shown that the software can be adapted for the analysis of MHC-like molecules. Moving forward, this MHC analysis has the potential to classify the antigen binding properties of highly-divergent MHC sequences from a broad range of species, providing insights where phylogenetic approaches prove difficult. This software represents a strong addition to the existing toolkit for repertoire analysis of diverse molecular species.

## Materials and methods

### Key resources table

| Reagent type (species) or resource | Designation | Source or reference | Identifiers | Additional information |
|---|---|---|---|---|
| Software, algorithm | Jupyter notebook | DOI:10.3233/978-1-61499-649-1-87 | RRID:SCR_018413 | https://pypi.org/project/jupyter-client/5.2.3/ |
| Software, algorithm | MatPlotLib | DOI:10.1109/MCSE.2007.55 | RRID:SCR_008624 | http://matplotlib.sourceforge.net |
| Software, algorithm | Seaborn | DOI:10.5281/zenodo.12710 | RRID:SCR_018132 | https://seaborn.pydata.org/ |

*Continued on next page*

*Continued*

| Reagent type (species) or resource | Designation | Source or reference | Identifiers | Additional information |
|---|---|---|---|---|
| Software, algorithm | Pandas | https://github.com/pandas-dev/pandas | RRID:SCR_018214 | https://pandas.pydata.org |
| Software, algorithm | SciPy | DOI:10.1038/s41592-019-0686-2 | RRID:SCR_008058 | http://www.scipy.org/ |
| Software, algorithm | Scikit-Learn | DOI:10.5555/1953048.2078195 | RRID:SCR_002577 | http://scikit-learn.org/ |
| Software, algorithm | AIMS | This Paper | Boughter et al. 2020 | https://github.com/ctboughter/AIMS |
| Software, algorithm | TCRdist | *Dash et al., 2017* DOI:10.1038/nature22383 | | https://github.com/phbradley/tcr-dist |
| Software, algorithm | IgBLAST | DOI:10.1093/nar/gkt382 | RRID:SCR_002873 | http://www.ncbi.nlm.nih.gov/igblast/ |
| Software, algorithm | BLAST | DOI:10.1093/nar/gkv1290 | RRID:SCR_004870 | http://blast.ncbi.nlm.nih.gov/Blast.cgi |

## Software

All analyses were performed in python, with code tested and finalized using Jupyter Notebooks (*Kluyver et al., 2016*). Figures were generated with MatPlotLib (*Hunter, 2007*) or seaborn (*Ziegler et al., 2014*), while the majority of data analysis was carried out using Pandas (*McKinney, 2015*), SciPy (*Virtanen et al., 2020*), and SciKit-learn (*Pedregosa et al., 2011*). All code and data is available at https://github.com/ctboughter/AIMS, including the original Jupyter Notebooks used to generate the data in this manuscript as well as generalized Notebooks and a python-based GUI application for analysis of novel datasets (*Boughter, 2020*; copy archived at swh:1:rev:f6c855ef4a7ce63f72dba6b34e9d0e9edd9200ce).

## Statistical analysis

Error bars in all plots are provided by the standard deviation of 1000 bootstrap iterations. Statistical significance is calculated using either a two-sided nonparametric Studentized bootstrap or a two-sided nonparametric permutation test as outlined in 'Bootstrap Methods and Their Application' (*Davison and Hinkley, 2011*). For the Studentized bootstrap, the bootstrapped data are drawn from a resampling of the null distribution of the data, with replacement. Practically, this entails combining the polyreactive and non-polyreactive antibodies into a single matrix, without labels, and using the Scikit-learn resample module to randomly separate this matrix into two classes, preserving the number of sequences in each population. To calculate bootstrapped averages, we draw from the empirical rather than null distribution. Statistical significance is estimated by calculating the p-value using the relation:

$$p = \frac{1 + \sharp(z^2 \geq z_0^2)}{R + 1} \qquad (4)$$

Here, we calculate the p-value by counting the number of bootstrap iterations where $z^2$ is greater than or equal to $z_0^2$. $z^2$ and $z_0^2$ are Studentized test statistics taken from the null and empirical and distributions, respectively. $R$ is the number of times this bootstrapping process is repeated. The general form of $z$ is given by:

$$z = \frac{\bar{Y}_2 - \bar{Y}_1}{(\frac{\sigma_2^2}{n_2} - \frac{\sigma_1^1}{n_1})^{1/2}} \qquad (5)$$

where $\bar{Y}$ represents the bootstrapped sample mean of each population, σ is the bootstrapped sample standard deviation, and $n$ is the number of samples. Populations 1 and 2 in this case correspond to polyreactive and non-polyreactive antibodies. To calculate $z$ for the empirical distribution ($z_0$), all values correspond to the empirical rather than bootstrapped values.

To calculate p-values for differences in mutual information, the permutation test was used rather than the Studentized bootstrap. Here, the test statistic $t$ is set to a simple difference of means, and rather than sampling with replacement from the empirical or null distributions with replacement, we randomly permute the data into 'polyreactive' or 'non-polyreactive' bins. We then count the number

of permutations where the randomly permuted test statistic is greater than or equal to the empirical test statistic. This count then replaces the count (#) in the above equation for *p*.

## Acknowledgements

This work was supported by the National Science Foundation through grant MCB-1517221 (BR and CTB), NIH NIBIB Training Grant T32 EB009412 (CTB), and NIH NIAID R01 AI115471, R01 AI147954 (EJA and CTB). We especially want to thank Cyril Planchais, Hugo Mouquet, and Michel Nussenzweig, for their assistance in procuring many of the antibody sequences used in this work. CTB acknowledges the University of Chicago's Research Computing Center Midway resource, where some of these calculations were carried out, as well as Kristof Nolan, Caitlin Castro, Ryan Duncombe, and Nabil Faruk for their insightful discussions of the research.

## Additional information

### Funding

| Funder | Grant reference number | Author |
|---|---|---|
| National Institute of Biomedical Imaging and Bioengineering | EB009412 | Christopher T Boughter |
| National Institute of Allergy and Infectious Diseases | AI147954 | Christopher T Boughter<br>Marta T Borowska<br>Erin J Adams |
| National Institute of Allergy and Infectious Diseases | AI115471 | Christopher T Boughter<br>Marta T Borowska<br>Erin J Adams |
| National Science Foundation | MCB-1517221 | Christopher T Boughter<br>Benoit Roux |
| National Institute of Allergy and Infectious Diseases | AI125250 | Albert Bendelac |

The funders had no role in study design, data collection and interpretation, or the decision to submit the work for publication.

### Author contributions

Christopher T Boughter, Conceptualization, Resources, Data curation, Software, Formal analysis, Validation, Investigation, Visualization, Methodology, Writing - original draft, Writing - review and editing; Marta T Borowska, Conceptualization, Data curation, Writing - review and editing; Jenna J Guthmiller, Patrick C Wilson, Resources, Data curation, Writing - review and editing; Albert Bendelac, Conceptualization, Resources, Data curation, Writing - review and editing; Benoit Roux, Erin J Adams, Conceptualization, Resources, Data curation, Supervision, Funding acquisition, Project administration, Writing - review and editing

### Author ORCIDs

Christopher T Boughter https://orcid.org/0000-0002-7106-4699
Benoit Roux http://orcid.org/0000-0002-5254-2712
Erin J Adams https://orcid.org/0000-0002-6271-8574

### Decision letter and Author response

Decision letter https://doi.org/10.7554/eLife.61393.sa1
Author response https://doi.org/10.7554/eLife.61393.sa2

## Additional files

### Supplementary files

• Transparent reporting form

### Data availability

All data generated and all code used for analysis in this study has been published on GitHub at http://github.com/ctboughter/AIMS (copy archived at https://archive.softwareheritage.org/swh:1:rev:f6c855ef4a7ce63f72dba6b34e9d0e9edd9200ce/).

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

# Appendix 1

**Appendix 1—table 1.** List of all biophysical properties used for this study.
For hotspot detecting variables (HS) a simplified form of the description is used. For more in-depth descriptions, the original reference should be used.

| Property shorthand | Description |
| --- | --- |
| Phob1 | Hydrophobicity scale [−1,1] |
| Charge | Charge [ec] |
| Phob2 | Octanol-interface hydrophobicity scale |
| Bulk | Side-chain bulkiness |
| Flex | Side-chain flexibility |
| KD1 | Helix/bend preference |
| KD2 | Side-chain size |
| KD3 | Extended structure preference |
| KD4 | Hydrophobicity |
| KD5 | Double-bend preference |
| KD6 | Flat extended preference |
| KD7 | Partial specific volume |
| KD8 | Occurrence in alpha-region |
| KD9 | pK-C |
| KD10 | Surrounding hydrophobicity |
| HS1 | Normalized positional residue Freq at helix C-term |
| HS2 | Normalized positional residue Freq at helix C4-term |
| HS3 | Spin-spin coupling constants |
| HS4 | Random parameter |
| HS5 | pK-N |
| HS6 | Alpha-helix indices for beta-proteins |
| HS7 | Linker propensity from 2-linker dataset |
| HS8 | Linker propensity from long dataset |
| HS9 | Normalized relative Freq of helix end |
| HS10 | Normalized relative Freq of double bend |
| HS11 | pK-COOH |
| HS12 | Relative mutability |
| HS13 | Kerr-constant increments |
| HS14 | Net charge |
| HS15 | Norm Freq Zeta-R |
| HS16 | Hydropathy scale |
| HS17 | Ratio of average computed composition |
| HS18 | Intercept in regression analysis |
| HS19 | Correlation coefficient in Reg Anal |
| HS20 | Weights for alpha-helix at window pos |
| HS21 | Weights for beta-sheet at window pos −3 |
| HS22 | Weights for beta-sheet at window pos 3 |
| HS23 | Weights for coil at win pos −5 |

*Continued on next page*

*Appendix 1—table 1 continued*

| Property shorthand | Description |
| --- | --- |
| HS24 | Weights coil win pos −4 |
| HS25 | Weights coil win pos 6 |
| HS26 | Avg Rel Frac occur in AL |
| HS27 | Avg Rel Frac occur in EL |
| HS28 | Avg Rel Frac occur in A0 |
| HS29 | Rel Pref at N |
| HS30 | Rel Pref at N1 |
| HS31 | Rel Pref at N2 |
| HS32 | Rel Pref at C1 |
| HS33 | Rel Pref at C |
| HS34 | Information measure for extended without H-bond |
| HS35 | Information measure for C-term turn |
| HS36 | Loss of SC hydropathy by helix formation |
| HS37 | Principal component 4 (*Sneath, 1966*) |
| HS38 | Zimm-Bragg parameter |
| HS39 | Normalized Freq of ZetaR |
| HS40 | Rel Pop conformational state A |
| HS41 | Rel Pop conformational state C |
| HS42 | Electron-ion interaction potential |
| HS43 | Free energy change of epsI to epsEx |
| HS44 | Free energy change of alphaRI to alphaRH |
| HS45 | Hydrophobicity coeff |
| HS46 | Principal property value z3 (*Hellberg et al., 1987*) |

