## [Decision Letter]

Thank you for submitting your article "Biochemical Patterns of Antibody Polyreactivity Revealed Through a Bioinformatics-Based Analysis of CDR Loops" for consideration by *eLife*. Your article has been reviewed by three peer reviewers, and the evaluation has been overseen by a Reviewing Editor and Satyajit Rath as the Senior Editor. The following individuals involved in review of your submission have agreed to reveal their identity: Brian Baker (Reviewer #1); Bart Haynes (Reviewer #2).

The reviewers have discussed the reviews with one another and the Reviewing Editor has drafted this decision to help you prepare a revised submission.

As the editors have judged that your manuscript is of interest, but as described below, we suggest substantial revisions. We would like to draw your attention to changes in our revision policy that we have made in response to COVID-19 (https://elifesciences.org/articles/57162). First, because many researchers have temporarily lost access to the labs, we will give authors as much time as they need to submit revised manuscripts. We are also offering, if you choose, to post the manuscript to bioRxiv (if it is not already there) along with this decision letter and a formal designation that the manuscript is "in revision at *eLife*". Please let us know if you would like to pursue this option. (If your work is more suitable for medRxiv, you will need to post the preprint yourself, as the mechanisms for us to do so are still in development.)

Summary:

The authors present a bioinformatics pipeline to analyze a large aggregate data set of nearly 1,500 antibody sequences, seeking to uncover the biophysical underpinning of antibody polyreactivity – i.e., broad low-affinity binding to diverse epitopes. This is an important topic, relevant not only to Ab therapeutics, but to other assessments of immune repertoires and considerations of protein-protein interaction hubs in general. The unbiased informatics approach that is taken, largely separated from structural considerations, is a distinctive feature of the work, and an attractive aspect of the study. The authors identify two key determinants of polyreactivity: First, the binding interface tends to be neutral, i.e., being neither strongly hydrophobic nor hydrophilic, and lacking significant positive or negative charge. Second, crosstalk (measured by mutual information) between and within CDR loops increases in the heavy chain of polyreactive antibodies. The method of sequence alignment leaves out most structural information (except aligning by the center of each CDR loop) yet retaining the positional context.

The research goal of identifying biophysical properties that allow for promiscuity of a protein selected for strong binding to a particular target is of general interest and biomedical importance, and the approach of combining statistical analyses and information metrics is appealing. The lack of a clear smoking gun is an important finding, as it clearly illustrates the subtleties at play. A classifier is described at the end of the paper, showing promise for sequence-based predictions and applications to therapeutic Abs.

We believe the paper may be suitable for *eLife* with revisions to address the following concerns.

Essential revisions:

1) The reviewers have concerns about the binary distinction between polyreactive vs. non-polyreactive. The authors essentially sum up this concern in the third paragraph of the subsection “Systematic Determination of the Key Contributions to Polyreactivity”, as well as in the ninth paragraph of the Discussion. The fact that pharma has nonetheless articulated a test for polyreactivity does not make the problem binary, which is clearly recognized by the authors. One way around this would be to repeat the analyses in Figures 1-3 on the more stringently separated (albeit now smaller) dataset used for Figure 4. How do things change? Are there any stronger signals?

Also, this limitation, while articulated well in the Discussion, could be brought up in the Introduction. Forcing a non-binary problem to be binary is by no means unique to this study, but it is a limitation best addressed upfront. As mentioned above, a strength of the paper is the demonstration of the scope of the challenge.

2) A major concern is that, with the biased removal of antibodies that are harder to classify, the true statistics are distorted. This occurs in two places. First, to avoid overfitting of LDA, only input vectors with the largest average differences between the two populations are kept. But this necessarily introduces bias which simplifies the classification problem and distorts the statistical structure in the original data set. Second, upon observing that an intermediate exists between the two classes when applying LDA to a subsampled data set, the authors removed the antibodies that bind a moderate number of select ligands from the analysis. But this continuum of polyreactivity is a real, and likely important, data feature. Since all the subsequent analyses act on the input data, it is necessary to address, with a clear description and supporting evidence, that the main findings remain valid when using the full data set. After this is addressed, direct comparisons – LDA vs. PCA, and with other methods – should be provided to justify the choice of LDA over PCA, and the performance of this newly developed pipeline relative to existing methods.

3) There seems to be a jump from the LDA to mutual information, then back to LDA in the application to therapeutic antibodies. While the physical hints the mutual information gives as mentioned in the Discussion is appreciated, and this can be built on, is this incorporated at all into the analysis of the therapeutic antibodies?

4) Related to point 3, it is not clear why mutual information is a good choice for detecting correlated changes between residues, and there is no comparison to other methods and previous studies. Moreover, mutual information won't be able to distinguish between coevolution and crosstalk between residues. For instance, Direct Coupling Analysis (DCA) indicates that coevolving residues are often in physical proximity. This points to a weakness of lacking the structural information in this work. Thus, it is unclear whether increased mutual information really indicates crosstalk, or instead signals coevolution between residues. This ambiguity undermines the second determinant of polyreactivity.

5) The analysis is finally applied to therapeutic antibodies, which is considered a logical next step. But this is not well motivated – why, in the first place, shall one expect correlations between polyreactivity status of naturally-derived antibodies and the acceptance or discontinuation of a therapeutic antibody? Hence, this section didn't lend validation support to the approach. Perhaps the manuscript could be shortened by deletion of the therapeutic dataset. These antibodies are not natural and it is not clear what approved versus discontinued antibody means.

6) There are other major omissions. First, no performance comparison between this new software and other available methods of antibody sequence analysis is presented (all is said in words, with no supporting figures). Second, it is stated that the approach has been adapted and applied to analyze MHC-like molecules, but no data or references are provided.

7) While in general the manuscript is quite clear, some sections are less so. Is there nothing to be discerned from the identity of the top 10 weights shown in Figure 4? We are only given the identity of three (subsection “Systematic Determination of the Key Contributions to Polyreactivity”, last paragraph), and even then these are only mentioned in a cursory manner. How do these and the other heavily weighted terms relate to the biophysical clues gleaned from the earlier analyses?

8) In the Abstract, the word "offensive" is not defined and would best be "neutral charge" for reader understanding.

9) Introduction, last paragraph, it is not clear in the first sentence if there are 1500 polyreactive abs or a total of 1500 abs studied. It is clarified later but not clear here.

10) The authors use a relatively small panel of antigens to define polyreactivity, whereas a chip type of assay with more proteins may be more precise. This might be mentioned as a way to make the discrimination between PR and non-PR abs more precise.

11) Figure 1 is terrific and very interesting. The fact that *V_H_*1-69 in over-represented in the polyreactive group of antibodies is important, both because of the propensity of polyreactive/rheumatoid factor B cells in fetal liver to be *V_H_*1-69 and because the polar region (PR)-distal HIV gp41 MPER Abs frequently use *V_H_*1-69. Perhaps these associations could be referenced and discussed.

12) In the fifth paragraph of the subsection “A Surface-Level Analysis of Polyreactive Antibody Sequences”, can statistics be applied to determine if any difference between 27% and 17%?

13) Figure 3, were these p values corrected for multiple tests?

---

## [Author Response]

Essential revisions:1) The reviewers have concerns about the binary distinction between polyreactive vs. non-polyreactive. The authors essentially sum up this concern in the third paragraph of the subsection “Systematic Determination of the Key Contributions to Polyreactivity”, as well as in the ninth paragraph of the Discussion. The fact that pharma has nonetheless articulated a test for polyreactivity does not make the problem binary, which is clearly recognized by the authors. One way around this would be to repeat the analyses in Figure 1-3 on the more stringently separated (albeit now smaller) dataset used for Figure 4. How do things change? Are there any stronger signals?

We appreciate the reviewers’ cognizance of the difficult nuances inherent to the current definitions of polyreactivity available to us and the broader field. We agree that including the analysis of Figures 1-3 applied to the parsed dataset would help contextualize our findings, and these data have been added to the supplement (Figure 1—figure supplement 1, Figure 2—figure supplement 1, and Figure 3—figure supplement 1). In these new figures, we see stronger signals in the same direction as the general trends found in the analysis of the full dataset. These stronger signals are discussed across multiple sections (subsections “A Surface-Level Analysis of Polyreactive Antibody Sequences” and “A Position Sensitive Matrix Representation of Sequences Provides Further Insights into Polyreactivity”). An interesting difference can be seen in the mutual information analysis, where loop crosstalk is increased in all loops, not just those of the heavy chain. This is highlighted in newly added text in the subsection “An Information Theoretic Approach” and Figure 5—figure supplement 1. Now that the parsed dataset is discussed earlier in the manuscript, a description of the parsed dataset has been relocated to the subsection “Database”.

Also, this limitation, while articulated well in the Discussion, could be brought up in the Introduction. Forcing a non-binary problem to be binary is by no means unique to this study, but it is a limitation best addressed upfront. As mentioned above, a strength of the paper is the demonstration of the scope of the challenge.

This difficulty of reframing a non-binary problem as binary is now discussed in the Introduction, in addition to the pre-existing statements found in the Results section and Discussion section. This updated section of the Introduction can be found in the seventh paragraph, when discussing the many difficulties of the systemic characterization of polyreactive antibodies

2) A major concern is that, with the biased removal of antibodies that are harder to classify, the true statistics are distorted. This occurs in two places. First, to avoid overfitting of LDA, only input vectors with the largest average differences between the two populations are kept. But this necessarily introduces bias which simplifies the classification problem and distorts the statistical structure in the original data set.

First, to hopefully alleviate some concerns for the reviewers, we have included Figure 4—figure supplement 2, using LDA to classify the full dataset, removing the identified bias. While performance clearly suffers, we are still able to successfully identify antibody polyreactivity with 71% accuracy, or a 65% accuracy when using LDA as a canonical classifier.

Second, when discussing LDA in this manuscript, it is important to differentiate between the two distinct modes used. In mode 1, our goal is explicitly to find the vectors which best discriminate between polyreactive and non-polyreactive antibodies. So here the “bias” is simply a search method. However, we can show that using PCA to reduce dimensionality rather than the maximum difference algorithm results in only a slight decrease in the reported accuracies (shown in Author response image 1).

In mode 2, the more canonical classification mode, this concern of simplifying the classification problem would be valid if the selection of the vectors happened before the splitting of data into “test” and “training” sets. However, this step occurs once the data has already been split. In other words, for each replica of the classification testing, the vectors selected with the largest average differences are changing, albeit slightly. Conceptually, this is no different from using other approaches to reduce the dimensionality of the system. Rather than using PCA, which identifies the axes of maximal variance using a linear combination of the input vectors, we are identifying axes of maximal difference.

The strength of this approach is that the resultant classifier is a linear combination of initial input vectors, rather than a linear combination of vectors that are linear combinations of the input vectors. This increased clarity makes the classifier more interpretable.

**Author response image 1. respfig1:** To address reviewer comment #2, we regenerated the analysis from Figure 4A using PCA rather than the maximum difference algorithm to reduce dimensionality. While performance is modestly decreased across all datasets, we still see strong separation between polyreactive and non-polyreactive antibodies.

Second, upon observing that an intermediate exists between the two classes when applying LDA to a subsampled data set, the authors removed the antibodies that bind a moderate number of select ligands from the analysis. But this continuum of polyreactivity is a real, and likely important, data feature. Since all the subsequent analyses act on the input data, it is necessary to address, with a clear description and supporting evidence, that the main findings remain valid when using the full data set. After this is addressed, direct comparisons – LDA vs. PCA, and with other methods – should be provided to justify the choice of LDA over PCA, and the performance of this newly developed pipeline relative to existing methods.

It should be pointed out, the full dataset is used for the majority of the analysis in the paper (Figures 1-3, Figure 5). Please see the subsection “Database”, for this clarification. As was addressed in reviewer comment 1, these analyses have now been applied to the parsed datasets and show stronger effects.

Likewise, the use of PCA was shown to be ineffective in separating the parsed data in Figure 4—figure supplement 1. For completeness, the PCA analysis of the full dataset is now also included in Figure 4—figure supplement 2. PCA, whether applied to the parsed or full dataset, is incapable of separating the two classes.

Lastly, to our knowledge, no methods to discriminate polyreactive and non-polyreactive antibodies in a systematic way currently exist.

3) There seems to be a jump from the LDA to mutual information, then back to LDA in the application to therapeutic antibodies. While the physical hints the mutual information gives as mentioned in the Discussion is appreciated, and this can be built on, is this incorporated at all into the analysis of the therapeutic antibodies?

As addressed in comment 5, we have removed the therapeutic antibody section, helping improve the flow of the manuscript.

4) Related to point 3, it is not clear why mutual information is a good choice for detecting correlated changes between residues, and there is no comparison to other methods and previous studies. Moreover, mutual information won't be able to distinguish between coevolution and crosstalk between residues. For instance, Direct Coupling Analysis (DCA) indicates that coevolving residues are often in physical proximity. This points to a weakness of lacking the structural information in this work. Thus, it is unclear whether increased mutual information really indicates crosstalk, or instead signals coevolution between residues. This ambiguity undermines the second determinant of polyreactivity.

One of the key novel approaches in our manuscript is the quantification of antibody sequences using position sensitive Shannon Entropy and Mutual Information. While Shannon Entropy has been used previously in repertoire analysis, notably in the generation of “sequence logos” [Schneider and Stephens, Nucleic Acids Research 1990], analysis using Mutual Information has been less common. However, this lack of previous use does not preclude mutual information from being a useful tool. The novelty here is part of the strength of our manuscript.

The reviewers point regarding mutual information’s inability to determine whether the signal is originating from coevolved residues and direct crosstalk is correct, and in fact this is mentioned in the sixth paragraph of the Discussion. While direct coupling analysis may find that coevolving residues are often in physical proximity, that does not explain the relative increase in the mutual information for polyreactive antibodies. There is no reason to believe, a priori, that mutual information should be increased across and within the loops of polyreactive antibodies compared to those of non-polyreactive antibodies. Regardless of whether this increase in signal is due to coevolution throughout the process of VDJ recombination and somatic hypermutation or direct contact between side chains, the signal is stronger in polyreactive antibodies. Importantly, this signal becomes even more pronounced when analyzing the parsed dataset (Figure 5—figure supplement 1). Hopefully, future studies can address this using structural and biochemical approaches, but these are outside of the scope of the present manuscript.

5) The analysis is finally applied to therapeutic antibodies, which is considered a logical next step. But this is not well motivated – why, in the first place, shall one expect correlations between polyreactivity status of naturally-derived antibodies and the acceptance or discontinuation of a therapeutic antibody? Hence, this section didn't lend validation support to the approach. Perhaps the manuscript could be shortened by deletion of the therapeutic dataset. These antibodies are not natural and it is not clear what approved versus discontinued antibody means.

As pointed out by the reviewers, while polyreactivity can play an important role in the success of an antibody therapeutic, the precise link between natural polyreactivity outlined in this manuscript and polyreactivity in engineered antibodies is not necessarily clear. Indeed, the data from the original Supplementary Figure 8 (now removed) seems to suggest there is little to no correlation between the two.

The original goal of this section was to give an example of the flexibility of our approach and its utility as a general tool. However, tackling a problem as complex as the success of or failure of an antibody therapeutic in clinical trials was likely overly ambitious. As suggested by the reviewers, we have removed this section. We now exhibit the flexibility of our approach with the new MHC and MHC-like analysis section.

6) There are other major omissions. First, no performance comparison between this new software and other available methods of antibody sequence analysis is presented (all is said in words, with no supporting figures). Second, it is stated that the approach has been adapted and applied to analyze MHC-like molecules, but no data or references are provided.

To our knowledge, no other software or available methods for antibody sequence analysis exists that was either designed specifically for, or is flexible enough to include, the classification of polyreactive and non-polyreactive antibodies. While previous researchers have attempted to find patterns in polyreactivity (Introduction, sixth paragraph) none of these studies claim to be able to predict polyreactivity in antibodies from their studies. Other potential approaches (Introduction, seventh paragraph) have been attempted at some level, but likewise have no claims of prediction.

Those algorithms that have been generated for prediction primarily focus on structural considerations of antibody-antigen interactions; i.e. epitope prediction, binding interface prediction, and ligand identification [Graves et al. Antibodies 2020]. While in theory these approaches could be used to classify polyreactive antibodies, they were not designed with this application in mind.

Last, a manuscript that utilizes the application of the software to analyze MHC and MHC-like molecules is currently in preparation. However, to demonstrate the potential application, a simple test case has been outlined in a new section of the manuscript (subsection “Extension of the Analysis to MHC and MHC-Like Molecules”, Figure 6). Here we are tasked with the very simple analysis of separating peptide-binding and lipid-binding MHC/MHC-like molecules, and then classifying molecules identified in a recent publication. In addition, the GitHub page hosting this software has an example test case for the easy application of this analysis mode.

7) While in general the manuscript is quite clear, some sections are less so. Is there nothing to be discerned from the identity of the top 10 weights shown in Figure 4? We are only given the identity of three (subsection “Systematic Determination of the Key Contributions to Polyreactivity”, last paragraph), and even then these are only mentioned in a cursory manner. How do these and the other heavily weighted terms relate to the biophysical clues gleaned from the earlier analyses?

We appreciate the reviewer’s suggestion to more closely consider the conclusions from our analysis. These weights were not so strongly discussed in the original manuscript due to the speculative nature of some of our conclusions. Without a crystal structure of a polyreactive antibody, our conclusions can only be strung together using the bioinformatic results we have. Our revised discussion of these data can now be found in the eighth paragraph of the Discussion section.

8) In the Abstract, the word "offensive" is not defined and would best be "neutral charge" for reader understanding.

We thank the reviewer for this suggestion. This wording has been changed in the Abstract.

9) Introduction, last paragraph, it is not clear in the first sentence if there are 1500 polyreactive abs or a total of 1500 abs studied. It is clarified later but not clear here.

This point has now been clarified. Additionally, since the therapeutic antibodies are no longer considered in this manuscript, references to the number of antibodies in the study have been changed to “over 1000”.

10) The authors use a relatively small panel of antigens to define polyreactivity, whereas a chip type of assay with more proteins may be more precise. This might be mentioned as a way to make the discrimination between PR and non-PR abs more precise.

One major drawback to the currently available experimental data is the relatively limited scope of the antigens tested to determine polyreactivity. While this panel of 7 antigens is the standard in the field, a protein microarray of some sort, as is often used when testing for polyreactivity in therapeutic antibodies, would provide a more detailed picture of polyreactivity. A reference to this proposed improvement in the analysis of polyreactivity has been added to the tenth paragraph of the Discussion.

11) Figure 1 is terrific and very interesting. The fact that V_H_1-69 in over-represented in the polyreactive group of antibodies is important, both because of the propensity of polyreactive/rheumatoid factor B cells in fetal liver to be V_H_1-69 and because the polar region (PR)-distal HIV gp41 MPER Abs frequently use V_H_1-69. Perhaps these associations could be referenced and discussed.

We thank the reviewer for their appreciation of Figure 1, and for their additional important context on *V_H_*1-69, which we have now included. These and other conclusions regarding the results of Figure 1 can now be found in the second paragraph of the Discussion.

12) In the fifth paragraph of the subsection “A Surface-Level Analysis of Polyreactive Antibody Sequences”, can statistics be applied to determine if any difference between 27% and 17%?

While we agree with the reviewer that the application of statistics to these percentages would strengthen the findings, it is unclear how statistics could be applied here. Since Figure 1 and the data associated with it are direct reports of the “count” of some property, here the count of a specific gene, it is difficult to acquire a standard error for this metric. Ideally, future experiments may quantify the gene usage of polyreactive antibodies across individual mice or humans, allowing for a more precise understanding of the deviation in these values.

13) Figure 3, were these p values corrected for multiple tests?

Originally, the p-values of Figure 3 were not corrected for multiple tests. Figure 3A and B now contain p-values at a level determined by the Bonferroni correction. It should be noted, when applying the Bonferroni correction to the data of Figure 3C and D, all differences fail to satisfy the corrected p-value level. However, a known shortcoming of this correction is a tendency towards an overly conservative analysis of the data. In this case where the number of tested points is 104, we believe that the multiple test correction is not a proper correction to our statistics. For increased data transparency, we have included the data profiles of the null distributions used for statistical comparisons in Figure 3—figure supplement 2.